# "The United States of Lyncherdom": Humor and Outrage in Percival Everett's *The Trees* (2021)

Michel Feith [1,2]

1   Department of English Studies, Nantes University, 44300 Nantes, France; michel.feith@univ-nantes.fr
2   Center for Research on Nations, Identities and Interculturality, Nantes University, 44300 Nantes, France

**Abstract:** An oeuvre as redolent with the spirit of satire and humor as Percival Everett's can be said to represent, at the same time, an anthology of humorous devices—a "humorology," so to speak—and a self-reflexive meditation on the existential, philosophical and/or metaphysical implications of such an attitude to language and life. *The Trees* (2021) is a book about lynching, in which a series of gruesome murders all allude to the martyrdom of Emmett Till. Even though such subject matter seems antinomic to humor, the novel is rife with it. We propose an examination of the various guises of humor in this text, from wordplay and carnivalesque inversion to the more sinister *humour noir*, black or gallows humor, and an assessment of their dynamic modus operandi in relation to political satire, literary parody and the expression of the unconscious. The three axes of our analysis of the subversive strategies of the novel will be the poetics of naming, from parody to a form of sublime; the grotesque, macabre treatment of bodies; and the question of affect, the dual tonality of the novel vexingly conjugating the emotional distance and release of humor with a sense of outrage both toned down and exacerbated by ironic indirection. In keeping with the ethos of Menippean satire, humor is, therefore, both medium and message.

**Keywords:** Percival Everett; humor; satire; lynching; grotesque

## 1. Introduction

Humor is a key feature of Percival Everett's fiction and public persona. It is found abundantly in most of his novels and stories, ranging from random puns to extensive parodic subversion and the most elaborate hoaxes. The literary con game performed by Thelonious Monk Ellison in *Erasure* (Everett 2001)–*Fuck*, a fake "Black Experience Novel" which backfires against its author, is taken to the next level by the practical joke Everett himself intended to set up with Telephone (2020). This recent work was stealthily published in three different versions to sow confusion among the community of his readers, who would argue in the dark about unexplainable discrepancies at the most literal level. The COVID-19 pandemic disrupted the experiment, adding a level of irony to the original mischievous intent.

If "Everett's work [is] an array of challenges to be taken on by the reader in a dynamic relationship premised not on solution, resolution, explication, or explanation—all of which presuppose some sort of definitive endpoint—and focusing instead on the dynamic nature of the challenge itself" (Stewart 2020, p. 8), the recourse to humor is a central strategy to achieve this literary ethos. In its parodic, subversive, open-ended dimension, it debunks ready-made assumptions, categories and identities. Therefore, for critics to fail to take into account this jocular dimension of the work may be, if not a total, at least a tonal, mistake.[1] His latest novel, *The Trees* (Everett 2021), can easily be categorized as satire, but contrary to *American Desert* (Everett 2004) it does not content itself with an ironic criticism of the American Way of Life. *American Desert* follows the journey of Theodore Street—a man who has come back to life after dying in a car accident on his way to committing suicide—through the American West, ridiculing as it goes the consumer society, the media

circus, religious fanatics and mad Army scientists. *The Trees* is about lynching. The title alludes to the "strange fruit" of the eponymous song (*Trees* 223). The subject matter and the ironic treatment remind one of Mark Twain's scathingly sarcastic late satire, "The United States of Lyncherdom" (Twain 1901), but Everett's novel might be even more disturbing than Twain's outraged diatribe, due to the tonal mix that characterizes it, a discrepancy between the dramatic subject and the range of humor that filters the narration. The text juxtaposes gratuitous jokes and witticisms, grotesque situations, satire, caricature, sarcasm and outrage.

Even though the story of *The Trees* takes place in the contemporary United States, the eye of the plot is the lynching of Emmett Till in Money, Mississippi in 1955. This teenager from Chicago, on holiday in the South, was accused of disrespecting a White woman, Carolyn Bryant. Several nights later, her husband, Roy Bryant, and his half-brother J. W. Milam kidnapped, beat and shot Till in the head before throwing him into the Tallahatchie River weighted with a huge metal fan tied to his neck with barbed wire. The men were acquitted by a local, all-White jury, but later confessed their crime in a paid magazine interview. The outrage sparked by the case, heightened by the exposure, at his mother's behest, of Till's battered and bloated face in an open casket, was instrumental in furthering the Civil Rights campaign that led to the repeal of legal segregation in the South. Decades later, Carolyn Bryant allegedly admitted that Till had never been rude to her. The novel takes place seventy years after the murder. Three relatives of the lynchers are found dead, two in Money, one in Chicago, in ways that are reminiscent of the Emmett Till killing: faces beaten to a pulp, barbed wire tied around their necks, abundant blood from a knife wound (or a gunshot) found at the crime scene, bodies emasculated—a trait often found in lynchings, even though presumably not in the Till case (FBI Website 2022). Next to them is also found the cadaver of a disfigured African American man, holding in his hands the victims' testicles. Carolyn Bryant passes of a heart attack a few days later, frightened to death by the apparition of the corpse in her bedroom.[2] To deepen the mystery, the body of the Black man vanishes from the county morgue after each murder, only to be found again at the next crime scene.

The investigation is devolved to the Mississippi Bureau of Investigation, or MBI. Special Detectives Ed Morgan and Jim Davis are on the case, with the help of FBI agent Herberta Hind. All are African American, which creates tensions with the local residents and police: to quarrels over jurisdiction are added racial strains. If we may spoil the detective-story part of the plot, the murders are retaliation for the Till lynching by a group of Black activists, militant avengers who want to raise consciousness about racial injustice and strife. But there is a further twist, a jump into the irrational and fantastic: a contagion of similar crimes ensues, first in the form of seemingly unrelated copycat murders, then in attacks by throngs of zombie-like assailants who target racists or their descendants.

Can *The Trees* therefore be considered as going beyond satire, at least beyond the canonical definition of it? In his *Jesting in Earnest* (2019), Derek Maus places Everett's fictional project within the ambit of Menippean satire, a mode of writing that is both similar to and dissimilar from satire. The latter is classically defined as the derision of the general follies and foibles of humankind, or more pointed social criticism through the indirection of humor; its ethos is ironic, and the vantage point of the satirist is often one of splendid isolation and moral normativity. Menippean satire, for its part, is characterized by its Harlequin-like variety of literary forms and genres, attacks on various philosophical ideas and attitudes to life and strong hints at the limitations of human understanding (Maus 2019, p. 54). Feeding on the humorous free play of the intellect and literary pastiche, and closely related to Bakhtin's vision of both dialogism and carnivalesque subversion (p. 62), Menippean satire does not cultivate a fixed moral standpoint, placing the author's persona and the reader within the general flux and chaos of the world and text. The humor is both medium and message, a jocular debunking of philosophical or moral monologism. While its outlook often tends to be more inclusive than ironic, it is a complex form that comes in many tonal varieties.



*The Trees* grafts onto the satirical ethos of irony and the unambiguous condemnation of racial violence grotesque, parodic elements that seem to make light of the tragic dimension of the text. Maus describes such "tonal multiplicity" as a trait of Menippean satire, which comprises the element of "thematic heterogeneity" exemplified here, and whose effects range from the cacophonic clash of antithetical levels of discourse to the uncomfortable juxtaposition of opposite emotional responses (Maus 2019, p. 95). An analysis of the double simultaneous postulation of the novel, its uneasy mix of humor and outrage, urges us to conduct a few forays into the question of affect in literature, and especially in humor. Affect Theory tends to distinguish affect from emotion, even though it is difficult to fully disambiguate the two. While on one side of the spectrum Lawrence Grossberg almost divorces them—"emotion is the articulation of affect and ideology. Emotion is the ideological attempt to make sense of some affective production" (Grossberg 2010, p. 316)—a more classical approach keeps the connection, as for example, the Freudian notion of affect as the "subjective translation of an amount of energy pertaining to drives". While it can be divorced from representation, it is still related to emotions, whose substratum and psychic energy it provides. Due to this connection, affect is hardly ever unconscious (Laplanche and Pontalis [1967] 1976, p. 12). In his famous study of humor, Freud stated that laughter stems from the release of psychic energy cathected to the expectation of, or buildup to, an unpleasant or onerous emotion, such as fear, pity or guilt, often linked to the representation of some taboo or repressed content (Freud [1927] 1961, p. 162). It does have an affective dimension, but, as a discharge, can it be said to be itself an affect? Hilarity does feel like an affect and emotion: relief, pleasure, a form of joy or even "triumph". This pleasure need not be unadulterated, and the tonal ambiguity that Everett cultivates in his fiction is a key feature of transgressive humor. Humor might then best be described as a conversion of affect, from unpleasant expectation to pleasure, in the same way as mental tension is (hysterically?) converted into a physical symptom, the noisy heaving that is laughter.

In order to try and make sense of the types and functions of humor in this challenging text, we will favor three angles of attack, encapsulated in three words which, like acupuncture points, seem to radiate along the meridians of the text: names, bodies and affect. There is a poetics of naming in the novel, ranging from funny, caricatural patronyms to the sublimity of the names of the deceased. Bodies are often represented as mysterious, grotesque and the subject-matter of macabre humor. Humor and laughter are embodied as forms of subversion and resistance to ready-made narratives, stereotypes and exercises of power. If the emotional distance and release effected by humor amount to a conversion of affect, keeping outrage and horror at bay, cannot the reverse also be true, an awareness of trauma and outrage flashing back from the indirection of humor and parody? Cannot the final emphasis on murderous contagion also remind us of the contagion of affect, the communicativeness of laughter serving as its clearest illustration? Far from providing a reassurance, it seems that humor in *The Trees* is ambivalent and destabilizing, always shifting the ground beneath the reader's feet.

## 2. The Unbearable Lightness of Names

Names and naming are of primordial importance in the African American tradition, in connection with a diverse array of affects. Under slavery, names were imposed by the owners; the resulting loss of African names and connection with African identities bespoke an alienation to the master's power to define. In a similar way, racial stereotyping and name-calling are acts of ideological designation and alienation. Un-naming, renaming oneself, as many slaves did after Emancipation, or signifying on given names and clichés, can be considered acts of empowerment and/or subversion (Benston 1982, p. 5). In Black literature, and especially in Everett's *oeuvre*, this comes to overdetermine one of the cardinal functions of personal names in fiction: the "reality effect" accruing from the presence of common-looking or historical names in an otherwise imaginary world has been one of the main tools of the realistic novel since the eighteenth century, allowing for a degree of suspension of disbelief (Watt 1957, p. 22). A third dimension of naming is featured in *The*

*Trees*: the imposition of toponyms on the land as a symbolic correlative of the colonization of the continent by predominantly White settlers.

The novel's opening sentence is a satirical gloss on the *locus dramatis*: "Money, Mississippi looks exactly like it sounds. Named in that persistent Southern tradition of irony and with the attendant tradition of nescience, the name becomes slightly sad, a marker of self-conscious ignorance that might as well be embraced because, let's face it, it isn't going away" (*Trees* 5). The name of the town reads like a particularly crude phrasing of the American Dream, in all its crass greed and materialism, whose offspring has historically been both rogue capitalism and slavery, without any idealistic sugar-coating. If American toponyms often express the founders' hopes, the ironic contrast with the dirt-poor, half-illiterate hick town that is described in the rest of the novel could not be greater. Irony is a question of point of view: conveyed by the third-person narrator, and seemingly unnoticed by the inhabitants, the paradox of a "self-conscious ignorance" alludes to some repression rather than mere obliviousness and denies the community the benefit of innocence. The debunking ridicule of this incipit is immediately undercut by the tragic undertones linked to the historical memory of the lynching of Emmett Till, adding another, more sarcastic level of irony. These undertones also taint the farcical names given to the suburbs and streets of Money, "Small Change" (p. 5) and "Dime Drive" (p. 66), derisive coinages all.

This streak of funny, ridiculous names is pursued with the victims' monikers, Wheat Bryant, Junior Junior Milam and Granny C, Carolyn Bryant's nickname. The local sheriff is Red Jetty, the medical examiner is called Doctor Reverend Cad Fondle and his assistant is Jethro Tull (recalling the inventor of the sowing machine). The sheriff's deputies are Delroy Digby and Braden Brady, and the town diner is called Dinah for illiteracy (p. 38). The Black FBI agent in charge, Herberta Hind, is nicknamed Herbie Hind, a mildly lewd suggestion, whereas three Asian American police men and women met in California are called Ho, Chi and Minh (p. 211). These names cover a whole array of onomastic humor corresponding to some of the main themes of humor in general, allowing one to draw a crude typology of humor, or "humorology". Wheat, Jetty and Tull, referring to concrete, prosaic elements, seem to allude to the literal, boorish mores of the rural community of Money. The doubling of the initial consonants in the names of the sheriff's assistants, rather than a reminiscence of Marvel superheroes, may point to intellectual stuttering and lack of imagination. The names tie in with the disparaging caricature of redneck country that is obtained throughout the novel. Critics have noticed that Everett usually defuses regional stereotypes about the West and South (Stewart 2020, p. 173), but here the satire is ruthless. This seems to align with Thomas Hobbes's view of the main function of humor as the disparagement of the Other and the creation of a feeling of superiority (Hobbes [1651] 1997, p. 52). It also bolsters the traditional rule-of-thumb distinction between humor and irony—the former is to laugh *with* another, whereas the latter is to laugh *at* the other—providing a first glimpse into the affects or emotions involved in hilarity.

The satiric, disparaging humor also targets "serious" institutions like the police and the church, with a critical or subversive intent. The policemen's stupidity and their prejudices come under attack: in addition to the local force from Money, there is a brief but intense portrait of a jingoistic, racist FBI veteran from Texas called Hickory Spit, who dies of a stroke after delivering a fevered rant about the next race war (p. 281): an act of poetic justice? Reverend Cad Fondle's name may imply a possible allusion to the sex scandals recently plaguing religious leaders, even if Fondle's flaw is the KKK version of white supremacy rather than pedophilia. Humor broaches taboo subjects, like sexuality, but jokes about the latter are featured only scarcely in *The Trees*, as in the mild pun about Herbie Hind. Puns hinge on the polysemy of phonetic chains, both the *double entendre* of ambiguous words and the semantic possibilities of alternative parsing. In funny or arbitrary names one can also find the pleasure of playing with the materiality of language through repetitions, alliterations or tongue-twisters. This foregrounding of the materiality and ambiguities of language refers to the self-reflexive dimension of humor, concentrated in an onomastic

form in the name of the English-born FBI medical examiner Helvetica Quip, pairing a typographic font and a witticism (p. 124).

The punning, arbitrary or allegorical names breach the "reality effect", precluding both referential illusion and reader identification. As aesthetic distancing devices pointing to the artificiality of the fiction, they also create emotional distance. This seems to tie in with Bergson's view that "the comic demands something like a momentary anesthesia of the heart. Its appeal is to intelligence, pure and simple" (Bergson [1900] 1914, p. 5). In *American Desert*, similar devices create satirical effect and encourage the reader to question the ideological nature of contemporary U.S. society. In *The Trees*, it fulfils the additional function of comic relief, of emotional respite from the represented social world of oppressive, sweltering racism and violence, overshadowed by the lynching of Emmett Till.

Names have such symbolic power, argues Judith Butler, because they are always given by another within a social context, which is also a context of power. They reveal our "linguistic vulnerability" (Butler 1997, p. 1), the powers of attribution and placement of these performative utterances. "One is, as it were, brought into social location and time through being named. And one is dependent upon another for one's name, for the designation that is supposed to confer singularity" (p. 30). This is presumably why insults are also called "name-calling," and why hate speech is also an alienating bestowing of reified, despised identities, evoking feelings of shame or humiliation. Humor can participate in this alienation, as in the case of racist or sexist jokes, but it may also defuse it, a process that the novel illustrates repeatedly. There are several amusing passages when the people of Money stumble on their own racism in the presence of the MBI and FBI: "The ni—the Black man is missing" (*Trees* 51). Or quibbles between the sheriff and the agents about their title of "special detectives":

"Special detectives," Jetty repeated.

"And that's not just because we're Black", Jim said. "Though plenty true because we are". (p. 32)

This little bout revolves around the polysemy of the word "special," which may mean cognitively challenged or exceptional. By calling out the sheriff's assumed prejudice, Jim turns the tables on him, demonstrating his own wit and converting an insult into a statement of pride. In another occurrence, the sheriff calls the agents "smart-asses" and Jim, addressing his colleague, replies: "He wanted to say *uppity*. Didn't he want to say *uppity*? I could hear it even though he didn't say it" (p. 34). The word "uppity" used to refer to a pretentious Black person, one who did not know their place, and it could carry threats of lynching. By teasing out the implicit insult, Jim challenges the sheriff, an attitude that is made possible only by his hierarchical position within law enforcement. Yet, this position of power is not without its contradictions, and the Black MBI and FBI agents feel qualms about their participation in a White-dominated power structure. For example, Herberta Hind's father did not approve of his daughter's job because he distrusted White people; yet he himself worked in the White House: "Even he liked the irony" (p. 163). Wit appears as a way to keep one's sanity under pressure. Humor may be a catalyst for racial and social tensions, but it might also help negotiate them and, through strategies of indirection and skirmish, keep the violence within language.

These examples illustrate some of the strategies of African American humor in a racially charged context, especially the various meanings attributed to the trope of "signifying", a form of verbal art based on wit and indirection. In the vernacular speech community, signifying implies that the dictionary meanings of words are not enough, and the need for interpretation arises when a comment, usually negative, is made either through metaphorical displacement or by addressing a person other than the target (Mitchell-Kernan [1973] 1990, p. 311). Playfulness and aggression coexist in a complex affective mix: the result of the exchange might be physical attack, but on most occasions the humor and virtuosity of the wordplay defuse these potentialities. Part of the pleasure of the text is the enjoyment of Ed and Jim's humorous badass assertiveness, in which they

seem to be intertextual tributes to Chester Himes's NYPD maverick detectives Coffin Ed Johnson and Gravedigger Jones (Himes [1964] 1998). Here, signifying takes on the literary meaning given it by Henry Louis Gates, Jr., as critical parody and revision, both within the body of Black literature and in relation to mainstream discourse and culture (Gates 1988, p. xxiii). *The Trees* signifies, among other aspects, on the genres of the detective story and zombie literature, and on white supremacist ideology. Even though signifying is only one of the strategies deployed in Everett's Menippean satire, and its racial focus is one that he regularly rejects as limiting (Maus 2019, p. 161), it is especially relevant in a novel about lynching that he should sound the blackness of the joke.

In addition to the comic aspect of onomastics, names and naming in *The Trees*, starting with the tribute to Emmett Till, are also linked with tragedy, and even a form of sublimity. As a matter of fact, the initial band of avengers are led by a 105-year-old woman called Mama Z, who has compiled a file on each lynching taking place in the United States since 1913, the year of her birth and of her father's racist murder. Ironically, the man's name was Julius Lynch (p. 158), and Mama Z's real name is Adelaide Lynch. The allegorical dimension of contrived names becomes symptomatologic. The group invites a Black scholar, Damon Thruff, to compose a pamphlet intended to publicize the motive for their actions, to bear witness to the persistence of racially motivated crimes. The irony, of course, is that their revenge itself amounts to a spate of racially motivated crimes, since the descendants of the Till lynchers, albeit boorish and racist, are not suspected of being killers themselves. After reading the files, Thruff feels an urge to write down the names of all the victims individually, to counteract the specific perniciousness of lynching. "He realized that the similarity of their deaths had caused these men and women to be at once erased and coalesced like one piece, like one body. They were all numbers and no number at all, many and one, a symptom, a sign" (p. 171). This caricatural version of the national motto *E Pluribus Unum* is based on a perverse form of erasure and invisibility: the serial, stereotyped perpetration erases the individuality of the victims, even in the very act of memorializing them. Lynching is the symptom of a national disease, described like the rituals of a serial killer. An inquiry into crimes potentially committed by such a psychopath ends up revealing that the nation as a whole can be envisaged as a collective serial killer.

Thruff copies each name with a pencil to endow them once more with a fleeting existence: "When I write the names they become real, not just statistics. When I write the names they become real again. It's almost like they get a few more seconds here [...] I would never be able to make up this many names. The names have to be real. They have to be real. Don't they? [...] When I'm done, I'm going to erase every name, set them free". (p. 190)

This is "ghost writing". Penning the names gives them existential depth, the weight of attention and meditation. It creates a spectral double of the victims. Like ghosts in folklore, the haunting hints at unfinished business—in this case, the exposure of an injustice—and must end in a dissolution that releases the soul for its passing.[3] Transiency is not the only mode of inscription of the names in the novel: a long litany of victims takes up ten pages (pp. 185–94). The list also features Chinese names, and Italian- or Spanish-sounding ones, to bring home the fact that racial violence in the U.S. is not only between Black and White. Emmett Till's name is included (p. 191), as is Philando Castile's (p. 194); the latter directly alludes to recent police violence, the Black Lives Matter movement and the attached Say Their Names campaign. These names represent the irruption of the real into the fictional work. They "have to be real": the text becomes a repository, a memorial for the victims. One is reminded of the Vietnam War memorial in Washington D.C., all the names chiseled in a marble wall for families and friends to draw pencil imprints on pieces of paper they can carry away. The "reality effect" is all the more striking because of the contrast with the absurd, arbitrary and clearly fictional names discussed earlier: they stand on a different ontological plane. The tribute names reverse the previously described gesture of emotional distancing through humor; they reinvest the names, through affective cathexis, with grief and mourning. This spiritual ritual of inscription, memorialization and sometimes erasure

mimics the grieving process. The feeling of absurdity is no less pervading in the reality of American violence than in the exaggerations of satire.

There is, in the novel, an "unbearable lightness" (Kundera 1984) of naming. Names encapsulate, to a certain degree, its ethos, an ambiguous, double-toned mixture of comedy and tragedy. The lightness of the funny names is part of the stinging social caricature, yet their ostentatious artificiality frustrates attempts at a realistic reading of a text dealing with dead-serious issues. On the other hand, the list of victims ushers in a pathos stemming from the fact that these names were once attached to bodies, and are now powerless to express the suffering, injustice and outrage of racial violence—the impotence of language to express realities or feelings is one way of defining the sublime.

### 3. The Curious Case of the Vanishing Bodies: *Humour Noir*, Grotesque and the Evidential Paradigm

The dual tonality of the novel, identified in the treatment of names, is also found in its narrative core, the reverse, revenge lynching of White supremacists and their descendants. The quasi-surrealistic tribulations of corpses endow the tragic subject with a touch of *humour noir*, or dark humor, that addresses our mortality without deviating from the pressing political issues involved.

The murders are imitations, pastiches of the lynching of Emmett Till. As with the memorial list of the lynched, one could say that the original event belongs to a different ontological sphere, and that due reverence to the grief and outrage is preserved. The simulacra, on the other hand, in spite of their serious intent, structurally function like practical jokes. They take the form of riddles, a Black man with a battered face holding the testicles of a lynched White man. Remarkably, nobody identifies the allusions at first, not even the African American law enforcement agents. The *mise-en-scène* is a grotesque display, with all the affective ambiguities of the grotesque. On the one hand, the mangling of the bodies has all the characteristics of abjection. According to Julia Kristeva, the cadaver is the ultimate figure of abjection, for the threat it symbolically poses both to life and to the categories that define any given culture's worldview: "A decaying body, lifeless, completely turned into dejection, blurred between the inanimate and the inorganic, a transitional swarming, inseparable lining of a human nature whose life is undistinguishable from the symbolic—the corpse represents fundamental pollution" (Kristeva [1980] 1982, p. 109). This metamorphic blurring of boundaries is one of the key elements of the grotesque (Bakhtin [1965] 1984, p. 24). It provokes a reaction of horror in the Gothic sense of paralysis and revulsion. As a matter of fact, Carolyn Bryant, the only character who immediately identified the connection with Emmett Till, dies of fear and guilt when exposed to the Black corpse brought to her bedroom under the cover of night (p. 107). The more humorous dimension of this grotesque originates in its derivative nature, its recourse to some of the devices of signifying or parody: "repetition with a difference" (Gates 1988, vol. xvii) or, in this case, crime rhyme. The rhetoric is one of inversion and displacement, since the Black man holds the White man's testicles in a reversal of the frequent castration of African American men by lynch mobs. The spiriting away of the Black man's body, to be found again at another crime scene, introduces an element of mystery. This may remind us of the similarities between the mechanics of dream logic and the rhetoric of wit, which both depend on the three main figures of condensation, displacement and indirection (Freud [1905] 1960, p. 166), to which we may add inversion and revision. The bafflement of the town confirms the dream-like, irrational appearance of the events. They are even more susceptible to an interpretation along the lines of the Bakhtinian grotesque, characterized by a subversive inversion of hierarchies and a focus on the materiality of the body. But rather than Bakhtin's vision of the carnivalesque as life-affirming comedy (Bakhtin [1965] 1984, p. 20), the grotesque treatment of the body in *The Trees* seems to fall within the ambit of gallows humor, dark humor or, as the French call it, *humour noir*. To us, "black humor" is here overdetermined by a racial component.[4]

The surrealist poet André Breton compiled a celebrated anthology of *humour noir*, which he thought contained subversive elements congruent with the surrealistic project: an attraction to taboo subjects, like death and sexuality; a radical questioning of rationality and bourgeois morality; an avenue into the Unconscious; a spirit of provocation and satire; and a playful attitude towards language (Breton [1939] 1966, p. 178). Breton also drew inspiration from Freud's economic view of humor as the relief of psychological tension. The surprise of humor undercuts the buildup to a painful emotion, thus sparing the psyche the corresponding expense of affect. It is often accompanied by a protective shift to the superego of the mental energies invested in a threatened ego. In spite of its seemingly cruel depictions, macabre humor allows us to confront our own mortality with less despair. There is pleasure gained in denying death and assuming the continuity of life, as in the following example of gallows humor, a quip by a condemned man walking to the scaffold: "Well, this week's beginning nicely" (Freud [1905] 1960, p. 229). Another aspect of macabre humor consists of desublimating the sacred aura of corpses, reducing them to the status of mere objects submitted to the laws of physics. This translation of the fear and reverence of death into the secular realm is another way of denying death's dominion, not by negating its reality, but by refusing its emotional contagion. This *memento mori* focuses on life.

One scene in *The Trees* perfectly exemplifies this form of macabre humor. Special detective Davis goes to visit a warehouse where unclaimed bodies are stored before being dispatched to medical schools all over the country. It is the point of origin of the Black corpse found with the White bodies in Money. The place has an industrial look: "It was like a cleaner's facility, except instead of shirts, blouses, and jackets, corpses, women and men, slid by on suspended rails. Farther away, through the center of the room, naked cadavers glided along, head to toe, on a conveyer belt" (*Trees* 197). In this body laundering outfit, one can discover some unexpected job descriptions, such as "nipple scrubber" or "ball washer" (p. 197). The employees may joke about their wares: "You kill'em, we chill'em"; "You stab'em, we slab'em"; "You slay'em, we lay'em" (p. 195). Others play soccer with a skull or play catch with an eyeball (p. 199). "Momentary anesthesia of the heart" indeed. The assembly—or disassembly—line goes further: it materializes the utter reification of the bodies, treated as raw materials rather than former persons. It satirically draws a supplementary connection between bodies and Money, suggesting that this lack of respect for dead people reflects the reification of living human beings, whether in lynching, formerly in slavery or in Capitalistic commodification, all in the name of power and greed. To the objection of potential sacrilege, the manager responds: "They're dead! Dead. Dead. Dead. Sacrilegious? Ain't no souls down there, just arms and legs, hands, heads, and elbows, tongues, testicles and nipples, ears and eyeballs. You need an eyeball, we ship you a fucking eyeball. It don't come with a nameplate or a testimonial. You just get an eyeball" (p. 199). The defensive, emotional tone, visible in the punctuation and repetitions, contradicts the man's professed indifference. The enumeration of body parts mimics the dismemberment of the corpses. The fact that this scene is situated just after the litany of names takes on special significance. In this world of the death-factory, the desublimation of macabre humor and the callous treatment of the bodies may amount to a survival strategy, a way to keep one's sanity in the overwhelming presence of death. Like the novel as a whole, this extract presents both a satirical indictment of contemporary American society and a veil of humor to cope with the emotional pressure of death and violence. African American humor is often endowed with such conjoined violence and criticism, which makes it truly "black humor". *The Trees* is definitely part of that tradition.[5]

According to Christelle Centi (Centi 2021, p. 145), Everett's novels waver between emotional desaturation due to an overabundance of the grotesque, as in *American Desert*, and an affective surfeit linked to the presence—or absence—of children's corpses, as in *The Water Cure* (Everett 2007) or *So Much Blue* (Everett 2017). *The Trees* mixes these two configurations within the same work, as it revolves around the murder of a young boy, Emmett Till, surrounded by a proliferation of the grotesque in the depiction of race relations in the United States.

Bodies in *The Trees* are also the centerpieces of the criminal investigation, placing the novel within Everett's own streak of revisited detective fictions. Corpses often vanish in these parodies, as in *Assumption* (Everett 2011) or *The Body of Martin Aguilera* (Everett 1994), which complexifies the gathering of post-mortem clues; even when they are not spirited away, they usually do not tell their secrets, thus defeating one of the key mechanisms of crime fiction, which historian Carlo Ginzburg called the "evidential paradigm". This paradigm is a model of interpretation based on the observation of small details and the use of rational deduction, leading to the resolution of mysteries. It is part of an epistemic mode of knowledge, elaborated in the second half of the nineteenth century, which encompasses more than detective fiction, since it also underwrites Freudian psychoanalysis, art criticism and the methods of various historical schools (Ginzburg [1986] 1989, p. 102). Christelle Centi places Everett's takes on deductive fiction within the category of the "metaphysical detective novels," which question the evidential paradigm and frustrate the reader by withholding the resolution of the enigma and the cathartic ending stemming from the construction of a fully explanatory narrative (Centi 2021, p. 113). Absent this cathartic resolution, the impression is that "the crime seems to be ongoing" (Dechêne 2018, p. 25), a very valid comment on the history of race-based hate crimes in the United States.

*The Trees* stages a to-and-fro motion between the deductive method of the traditional inquiry and the questioning of it. Rather than merely criticizing the evidential paradigm, the plot exceeds it. When opposed to the grotesque, irrational reactions of the White people, procedural rationality is condoned and upheld, but is then revealed to be lacking when a different irrationality is unleashed. Part of the humor and comedy of the novel proceeds from the people of Money's grotesque and ignorant reactions. They first tend to believe that the dead Black man must have killed the other victims, even though his own death remains mysterious. It seems that their brains are wired to attribute all crimes to Black people. When the body disappears, they believe he is a ghost, even though they carried him to the morgue themselves. One is reminded of the double meaning of "spook" in American slang, on which Ralph Ellison capitalized in the incipit to *Invisible Man* (Ellison [1952] 1965, p. 7). Dr. Rev. Fondle, medical examiner, preacher and KKK leader, offers a prize example of religious fundamentalist discourse: "O Gawd Jesus, I knows you have a plan, but us poor White mortals is scared to death down here with this strange nigger you keep sending. Is he an omen, oh Lawd, a sign, or is he the devil?" (p. 45). This parody features an ironic reversal of supremacist beliefs: the rationale for the spectral white robes and hoods worn by the Ku Klux Klan in their terror campaigns derived from an assumption of the superstition of Black people and their putative fear of ghosts. As multiple witness accounts tell, the victims were never fooled: they were terrorized by the violence, not by the costumes (McWhiney and Simkins [1973] 1990, p. 587). The pretended superiority of White people is constantly ridiculed, as when a cross-burning intended to chase them away is ignored by the Black detectives: "'I wish I had known,' Jim said, 'I forgot to be scared.'" (*Trees* 110). Another grotesque reaction of White fear, beyond the pale of Money, is that of President Donald J. Trump cowering under the Resolute desk when the White House is under attack, and being unable to come out because his obese body is stuck under it (p. 282).

Against this irrationalism, the evidential paradigm, as put in practice by the Black MBI and FBI agents, gives out results and leads to the resolution of the case. Mama Z's band of activists killed the members of the Bryant and Milam families, using as symbols and clues a few cadavers stolen from the corpse exchange. They were about to publish a pamphlet explaining their motives and meant to draw attention to the ongoing racial violence in the USA. But this Scooby-Doo plot is derailed when a spate of copycat lynchings of White people erupts all over the country, including in the aptly named—and real—Carbon County (p. 145). This looks more like a contagion than a causal relation, since the police did not publicize the relevant details of the murders. The final straw is reached when bands of presumably dead Black people, matted with dust and unkillable, attack White men, mostly targeting supremacists and racists; zombie, the nickname of the old root doctor, Mama

Z, becomes prophetic. It reads as though a generic contagion has occurred between the detective plot and the zombie genre. This "Walking Dead" finale is the culmination of the grotesque, parodic streak in the novel; the absurdity of this generic effraction is highly comical, but also highly ambiguous. What might appear as a jab at a fad, the repetitious surfeit of zombie movies, TV series, video games and books, may also suggest that the popularity of the genre reveals some nagging preoccupation in the national mind, possibly its political unconscious.

Zombies originally entered American popular culture at the time of the occupation of Haiti, in the 1920s; they represented the mysterious powers of Black Voodoo, bringing dead people back to life to be worked as slaves. They evoked racial Otherness until George A. Romero mutated the zombie into brain-dead creatures intent on eating other people's blood and gray matter, transmitting their condition like a disease through their bites (Coulombe 2012, p. 26). Roaming shopping centers and suburbia, they could be seen as a metaphor of deadening conformity. The resulting cultural compound is highly ambivalent. The zombie is a monster, and is therefore associated with the fear of the Other, but this Other can also be the self. It is doubly connected with trauma, as the victim of an attack that in turn becomes a threat (p. 64). It mostly symbolizes the uncanny, the familiar that morphs into a menacing Other and the return of the repressed, the sprouting of what lay buried underground, in the cemetery of the mind: death, the body, the social and racial Untouchables. At the height of the zombie insurrection in *The Trees*, the throngs single-mindedly shout one word, which is also the second, buried title of the novel, "Rise" (*Trees* 3). "*Rise*, it said. *Rise*. It left towns torn apart. Families grieved. Families assessed their histories. It was weather. Rise. It was a cloud. It was a front, a front of dead air" (p. 306). This apocalyptic imagery is couched in meteorological terms, but it is clearly a historical apocalypse, bringing to the fore the brutality of American racial power under the appearance of public and private respectability. One is reminded of the Weather Underground, a group of leftist activists of the 1970s, but most of all of the pervading fear of slave rebellions in the antebellum South. This blast from the past implies that all historical accounts have not been settled, and that in the age of Black Lives Matter there is a need for a new overthrow of White supremacy. From the jokes about the "spooky" nature of unexplainable events and racial Others to the parodic zombie uprising of the finale, *The Trees* portrays a "hauntology" (Derrida [1993] 1994, p. 51) of a country whose ontological reality is predicated on, and subverted by, the violence it represses.

This fantastic–allegorical trait of the narrative is taken up to a metafictional pitch at the very end of the novel, which takes place at Mama Z's house, and presents Damon Thruff typing the names of the lynched.

> "He's typing names", Mama Z said. "One name at a time. One name at a time. Every name".
>
> "Names", Ed said.
>
> "Shall I stop him?" Mama Z asked?
>
> Jim looked at Ed, then Hind. Gertrude was clearly confused. They were confused, yet not.
>
> "Shall I stop him?" the old woman asked again.
>
> Outside, in the distance, through the night air, the muffled cry came through, *Rise. Rise*.
>
> "Shall I stop him?" (p. 308)

This last question suggests that the army of the dead was raised by the act of writing. Damon becomes a conjure man, as hinted by his first name. This type of writerly magic is distinct from his earlier pencil inscriptions and erasure: rather than freeing the souls of the dead, he brings them back to exact revenge, mirroring his own outrage. This passage probes the powers and duties of fiction in conjuring up the memories of the victims, conjuring revolt and conjuring outrage. The ending of *The Trees* is suspended: no condemnation

of retaliatory violence is proffered, even though retaliatory racial violence is still racial violence, and does not overcome the root of the problem, the paradigm of race. Within the story world, the characters seem to have decided to let the rebellion run its course. Even though there is no illusion as to the power of fiction to induce change in the real world, given the urgency of the matter it seems that the mere denunciation of racial violence is not enough, and that this world deserves its apocalypse. "Realistic" or ethical truth is clouded by affective truth. The novel seems to exemplify another trait of the contemporary uses of Menippean satire, which Maus, after Weisenburger, calls "subversive" or "degenerative satire": it does not content itself with distrusting and delegitimizing totalizing worldviews and philosophical constructions, but acknowledges the complicity of narratives in their elaboration, and is led to metafictionally "sabotage" its own meaning-making process (Maus 2019, p. 60). Everett's novels disable the reader from adopting a comfortable moral posture, since they emphasize how influenced we have been by American ideologies, be they the narrative of Western expansion or the paradigm of race.

## 4. Affection and Infection: An Anatomy of Humor

The double-toned nature of *The Trees*, constantly hovering between comedy and tragedy, pathos and humor, pointing to the overlapping spaces and therefore to the limitations of the dichotomy between these two genres, draws attention to the affective dimension of literature and of humor. This reader's own uncomfortable impression that the humor of the text was often exacerbated by the violent, macabre backdrop made him wonder if the joke was not also on the reader. The deeply transgressive, politically incorrect side of Everett's humor catches us off guard, fostering complicity but not without a tinge of guilt, forcing us to confront our own ideologically conditioned reactions. In a pig Latin variation on the Roman satirist Horace's famous apophthegm De te fabula narratur (this story is about you), one could encode this impression as De te joculatio narratur. If humor is both the medium and the message, it seems to be due not only to its formal characteristics, but mostly to its emotional, or affective, impact, its perlocutory effect, to use Austin's typology. Contrary to the "illocutory" force of a speech act, *that is* its intentional purpose, like warning or supplication, the "perlocutory" force represents its effective impact on the addressee, such as persuasion, fright or seduction (Austin 1962, p. 101). Part of this effect has to do with affect.

The psychoanalytical interpretation of humor that we have so far mobilized, convincing as it is, may focus too much on the individual's psychosomatic response, whereas the taboos and moral or cognitive categories that jokes disturb belong to the social realm. According to Laure Flandrin, laughter is a stronglysocial emotion, and the sociology of laughter can reveal much about the place of the laugher within the collective. One may add that a discussion of humor can also enlighten the role of affects in the creation and maintenance of groups and their borders. Laughter is not only symptomatic; it is truly a socially performative act (Flandrin 2021, p. 179). Who you laugh with, who you laugh at and who does not laugh with you are community-building acts. Laughter can be inclusive, but it can just as well be exclusive, establishing links of solidarity between some at the expense of others. Racist or sexist jokes, like insults or name-calling, integrate the utterer in a community of like-minded people, synchronically (who is laughing?) and diachronically: "The speaker who utters the racial slur is thus citing that slur, making linguistic community with a history of speakers" (Butler 1997, p. 51). Flandrin's field study tends to show that, far from stemming from clearly established social positions and boundaries, racist jokes often target the group that is socially closest to the speaker and whose situational similarities threaten to erase the distinction in status. The performative dimension of humor is, here, an attempt to keep the Other at bay, trying to enforce boundaries rather than confirm them (Flandrin 2021, p. 197). The affects and emotions involved are, among others, a sense of reassurance and belonging, set against the degradation or humiliation of the Other. Even though her results concern France, and are not fully transposable to the environment of the United States, Flandrin herself draws a parallel with the "*angry white men* of deindustrial-

ized America" (p. 196). Within the context of Money, Mississippi, the age-old antagonism between the poor Whites and African Americans springs to mind.

Racial humor is shown to stem from the same affective sources as racial violence. The threat of confusion and the impossibility of keeping borders between groups air-tight is redoubled in the South by genetic insecurity. No-one is sure of being "pure" White, not even supremacists. Red Jetty, the Sheriff who, while not being shown as really intolerant, squabbled with the MBI agents by means of ironic, possibly racially charged innuendo, suddenly realizes that the Black man he had seen his father kill in the town street years before was actually his biological father (*Trees* 223–24). The CB handle of Charlotte, Wheat Bryant's wife, is "Hot Mamma Yella," due to the color of her trademark tank top, but to the Black MBI agents the meaning is radically different (p. 62). Who is to say that this joke does not reveal some buried genetic truth, or at least genetically based anxiety? Because they stem from the same social fears, humor might be able to defuse violence, as in the already-mentioned example of the joust over the term "special detectives" between Ed and Jim and the sheriff. Wit shifts group boundaries, either by calling out the sheriff's implied racism or, hopefully, by temporarily aligning him with his adversaries in the enjoyment of a clever word.

The carnivalesque ridiculing or subversive criticism of institutions like the police and the church also mobilize political group affiliations. The hilarious stump speech by then-President Trump at the end of the novel, a caricature straight from late-night talk shows, harnesses this most divisive and grotesque political figure to cement the community of readers against him, and so to define the text's implied readership. As a matter of fact, it is doubtful that Trump voters should read Everett's novels, especially this one. The speech is peppered with recognizable, infamous, yet funny sound bites: "my people tell me, they're the best people" or "no collusion" (p. 299). But what is pointed out is Trump's crass racism and bad faith: "But these niggers have gone off the rails and off the reservation, and they have to be stopped [...] I did not use the word *nigger*. I would never use the n-word. I'm the least racist person you will ever meet" (p. 300). This motivated, *ad hominem* satire is also a comment on the state of a nation that could elect such a clownish representative, venturing the hypothesis that race was a key factor in the vote. Beyond the archetypal humorous theme of the degradation of authority figures, this savage satirical caricature is also pervaded with anger and outrage.

Humor can, then, be a privileged angle to examine the dynamics of affect, in exact parallel with the treatment of outrage in the text. These dynamics may help account for one aspect of the novel that has remained mysterious to this reader: the connection between the realistic themes of the novel, racial prejudice and racial violence, and some elements that defy rationality and causality, like the lynching epidemic and the zombie invasion. The irrationality of these episodes may be considered as an allegory for the irrationality of race violence, or they might aim, in quasi-surrealistic fashion, at questioning rationality itself. Moreover, we might venture the hypothesis that, as textual effects, they echo the logic of some developments in Affect Theory.

Zombie invasions are usually imagined along epidemic lines, but in *The Trees* there are no instances of biting and metamorphosis. The contagion seems to have no other support than a shared, environmental sense of outrage, whose rationale is expressed by Mama Z's group, who feel they are meting out "retributive justice" and that they are soldiers in a low-intensity war (*Trees* 236)—before it spreads nationwide. One mechanism of textual contagion is duplication and serial replication. Its molecular expression is the comical stuttering of names with duplicated initials, like Delroy Digby and Braden Brady, which now takes on a more sinister connotation. The repetitive pattern of reverse lynchings in Money, followed by the disappearance and reappearance of the same Black body, takes on the aspect of "*comique de repetition*" or, more ominously, of what the French philosopher Henri Bergson considered as one of the mainsprings of humor, the imposition of mechanical patterns upon human processes or "something mechanical encrusted upon the living" (Bergson [1900] 1914, p. 49). In the copycat murders and zombie insurrection,

this mechanical iteration runs out of hand in a purely textual duplication that escapes material causality and the characters' intentionality. An earlier variation on this scheme is to be found in *American Desert* in the figures of the Christ clones manufactured by the Army in the hope of creating resurrecting soldiers (*Desert* 197). The conjunction of a mechanical (military–scientific) process, organic cell reproduction, and the grotesque—all the clones are somehow misshapen—parallels the textual contagion in *The Trees*. It spells out the sinister potentialities of the imposition of the mechanical over the human, whose homothetic counterpart could be the violent dehumanization and stereotypical behavior of serial killers.

The contagion and seriality in the novel seem to echo the formalization of affect in one of the branches of Affect Theory, the "noncognitivist" streak, envisioning affects as physical intensities that predate linguistic expression, and, therefore, both emotion and ideology (Houen 2020, p. 4). They can therefore serve as vectors of alienation or resistance. Some view affects as intensities circulating between bodies without necessarily obeying deterministic laws of causality or the directions of the conscious will (Seigworth and Gregg 2010, p. 1). For example, in a description of fear-mongering through the media, Brian Massumi defines a self-generating affective environment that shapes a new collective mind and identity. In one transitional, deterritorializing phase, the distinctions between subject and object are erased: "Prior to the distinction between agent and patient, in the bustle of the reawakening, there is no boundary yet between the body and its environment, or between the two of them and the correlated sign. Or between the dream and the event, or between all of the above and other bodies" (Massumi 2010, p. 66). Whereas one may remain skeptical as to the descriptive power of this model in the social realm, it seems to accurately describe the processes of *textual* contagion of the desire for revenge in the novel, from the copycat lynchings which seem to bear no causal connection to the original murders to the zombie insurrection. Humor is often said to be infectious when hilarity spreads throughout a room by means of involuntary mimetic influence. In *The Trees*, outrage is just as infectious, spreading like its twin along rhizomatic textual lines, between characters and between text and world, possibly alluding to the mysterious influences through which the literary work affects its readers.

In the "tonal multiplicity" of Everett's satire, outrage and humor may be indissociable, yet, if one had to venture a hypothesis, one would guess that in *The Trees* the baseline affect is outrage. Humor and irony echo the sense of absurdity stemming from grotesque forms of racial violence and racial prejudice. Aesthetically, they convey, as frequently throughout Everett's oeuvre, an indirect approach to emotion and express a refusal to indulge in pathos. The emotional distance thus achieved may avoid affective overload, both for the implied author and the reader. As a matter of fact, the topical nature of the novel—as an obvious response to the Trump presidency, which allowed some barely repressed expressions of racism to invade the public sphere again, and the Black Lives Matter movement that shed light on continuing police violence–hints at an affective context of anger and distress that was only compounded by the outbreak of another epidemic, COVID-19.

The conversion of affect that is the mechanism of humor may be seen as a survival strategy, a measure to keep one's sanity in the face of horror or, as Danielle Morgan puts it, "laughing to keep from dying" (Morgan 2020, p. 5). Yet, in the same way as the grotesque reverse lynchings in Money do not detract from the reverence due to Emmett Till's martyrdom—the etymological meaning of martyrdom is "bearing witness"—the dual tonality of the novel never stifles the outrage behind the veil of humor. If we compare *The Trees* to other novels picturing radical Black activist groups resorting to counter-terror through retaliatory reverse lynchings, such as Toni Morrison's *Song of Solomon* (1983) or John Edgar Wideman's *The Lynchers* (1973), Everett's novel refrains from condemning the use of such violence. Even though there are hints of the irony of fighting racial violence with racial violence, the ending is suspended and problematic: "Shall I stop him?" (*Trees* 308). The novel responds symmetrically to the White vengeance of imaginary wrongs, the pretext of rape to justify lynching, with the staging of an imaginary revenge, blown up

to apocalyptic proportions. Its stance on the powers and duty of literature also differs, at least superficially, from that of Chinese American writer Maxine Hong Kingston who, in her rewriting of the tale of Fa Mu Lan, the woman warrior, tried to articulate the story's revenge plot with her own ethics of non-violence by emphasizing the awareness-raising potential of literature: "The reporting is the vengeance--not the beheading, not the gutting, but the words" (Kingston [1975] 1989, p. 53). Beyond the horror, the satisfaction of "just deserts"—or "American deserts"—is another troubling affect that is somewhat permitted by the blatant fictionality of satire and humor. Humor once more allows for the expression of a taboo. If, as Elaine Scarry put it, torture—and what is lynching if not torture?—is "the unmaking of the world" and artistic creation a "world-making" reclaiming of language and agency (Scarry 1985, p. 22), can we say that language in *The Trees* is life-affirming? It seems to propose even more destruction, and the humor in particular exerts a form of linguistic violence. Yet, it is violence to counteract violence, by subverting the racist ideology that sustains it. The text does not provide us with easy solutions, but the very distance that is the prerogative of humor is key to disalienation from conditioned responses, be they racial hatred or the impulse to violent retaliation. One thought that might stem from the reading of the novel is that, due to the sheer enormity of the historical outrage in American history, one may wonder why there has not been more violence on the side of the oppressed, of the type portrayed in *The Trees*. The novel might spell out an indirect tribute to the forbearance of the victims and pose the disturbing question of how to deal with outrage and rage to prevent them from someday overwhelming the nation.

**Funding:** This research received no external funding.

**Conflicts of Interest:** The author declares no conflict of interest.

## Notes

[1] We could even coin a name for it, *Inflatism*, after the father of the main character Ralph in *Glyph* (Everett [1999] 2004), a poststructuralist university professor nicknamed Inflato by his son, who "was not a fat man, but he was bloated, moving as if he were larger than he actually was" (*Glyph* 6), a mediocre, conceited man "falling repeatedly into the same trap, the thought that he not only could talk about meaning, but that he could make it" (7). The figure of Inflato might be said to represent the academic "*esprit de sérieux*" of critics who take their own theories without a grain of salt or the necessary distance.

[2] Carolyn Bryant Donham died of natural causes between the release of *The Trees* and the publication of this article, on 25 April 2023, at the age of 88 (Fow 2023).

[3] This complex variation on erasure and invisibility prolongs Everett's probing of these issues in *Erasure* (Everett 2001), itself a tribute to Ralph Ellison's *Invisible Man*—it seems that in a violent, racist society, the fate of the dead mimics that of the living.

[4] According to Weisenburger, «Black Humor» was a loose category of dark fiction reaching "beyond satire," or beyond the traditional definition of satire, in the nineteen sixties and nineteen seventies. Its leading practitioners were Kurt Vonnegut, Joseph Heller, Thomas Pynchon, John Barth and Donald Barthelme. This production was characterized by very dark humor, absurdism, despair and self-referentiality. "Black Humor illustrates what happens in the postmodern novel when that fixed [reference] point [constitutive of irony] can no longer be trusted, when the 'subject' begins to dissolve and *humor* reveals it as yet another surface, or masking fiction, of a dissimulating humanity" (Weisenburger 1995, p. 101). While there is definitely a continuity between these writers' ethos and Everett's, the latter's relation to postmodernism is much more problematic, and the focus on issues of race complexifies the whole notion of "black" humor.

[5] "Not infrequently 'race humor' has a grim and even macabre quality. Such is the famous cartoon which appeared first in the *People's Voice* of New York after the Detroit riots. It portrayed two small white boys looking at hunting trophies hanging on the wall of father's den. Among them is the mounted head of a Negro. One small boy says proudly, 'Dad got that one in Detroit last week.'" (Burma [1973] 1990, p. 626).

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
