# Peer review of "“The United States of Lyncherdom”: Humor and Outrage in Percival Everett’s The Trees (2021)"

_humanities, doi:10.3390/h12050125_

Round 1

Reviewer 1 Report

A take on Everett's novel from the angle of humor studies is well taken, but the manuscript struggles to establish a clear argument and clear theoretical foundation. There is too much namedropping and not enough focus. It seems that at its core, it reads the novel as a postmodern text--which is in line with the importance of satire and humor in general. It makes valid and important analytical points, and certainly, names and bodies are important categories to investigate in the novel. Affect is an interesting lense, but its demarcation from humor (or its relation to it) are not entirely clear. Be bolder in your formulation of a clear thesis/argument that you keep in view throughout every subchapter.

The prose flows well overall, but there are some idiomatic issues where words are used in slightly different meanings.

Author Response

I have tried to focus my argument through revisions to the introduction and the third part. Some quotes have been deleted, so as to improve the ratio between analysis and theory.

Thank you very much for reviewing and helping me to improve my manuscript.

Reviewer 2 Report

There is much to like about this essay, which grapples with the disturbing and destabilizing dark humor of Everett's novel. The thesis is sound and, most of the time, well-argued. Yes, the essay often moves freely and loosely between the secondary literature on humor and the content of the novel, and does so in ways that often challenge the reader's ability to follow the flow of ideas, but I find this challenge to be suitable in an analysis of a novel that is equally loose and free-flowing (and destabilizing).  I have made minor comments throughout the manuscript: most of the early comments are affirmative of the argument, but in the final section of the essay I find that the essay gets a little too loose--meaning, I don't follow the logic of the argument in the final section as well as I did in the first two sections. The author needs to revisit this section and edit out paragraphs that don't transition well into each other. I think, in the end, this final section can be tightened and made more effective.

Overall, though, this is a compelling analysis of how Everett uses humor to engage with historical trauma and tragedy, and it adds something substantial to our understanding of his oeuvre, and the place of the Emmett Till lynching in African American memory.

Author Response

I have tried to streamline and focus my article, through modifications to the Introduction and the third part. The annotations to my original text have been especially useful, both for matters of style and to help me understand where the argument lacked clarity. Thank you vey much for the thorough reading.

Reviewer 3 Report

This is a well-written and original essay on Everett's The Trees that takes a productive approach to the novel and lays a useful groundwork for future critical commentary. The writing itself is very compelling and free of jargon without lacking in complexity, and there are only very few instances where the otherwise persistent coherence is broken a bit (the paragraph about “race humor” on page 8 is underdeveloped and seems arbitrary in its disconnectedness, as is the paragraph about Trump on page 9; I would also revise the paragraphs on contagion to combine them into a more straightforward argument).

Each of the two parts - one on humor, the other on affect, broadly speaking - is thorough and convincing both in its basic arguments and in their respective details. The reading of the novel is sound and perceptive, and the use of theory is wide-ranging and precise. Personally, I find Freud and Bergson to be rather tiresome reference points for discussions of humor by now, and especially the psychoanalytic approach could either be updated or scrapped for a more convincing or relevant one (as it happens later in the more sociological take), but this is a minor quibble, and the author does do interesting and valid things with Freud, so it's not a major issue, just food for thought.

I appreciate how nuanced the difficult discussion of literary names is, as reading them for their ambiguous meanings (or non-meanings) often is at risk of devolving into a highly speculative affair, as they necessarily need to be taken out of context to a certain extent - and if all meaning is created by context, this makes the meaning-making tricky. I think the author manages to steer clear of this for the most part, but there are some moments where they suggest what these names "may" (4) do, and that speculative direction is less convincing than the more context-dependent ones before (but again, this is a minor quibble).

Some minor typos/corrections: I would change "The title echoes the “strange fruits” of the eponymous song" (1) to "strange fruit" to match the actual title. The "Mississippi Bureau of investigation" should probably have the I capitalized. There's also some confusion about whether the author uses I or We as a pronoun, so this should be changed for consistency depending on how many authors there are.

While the essay is thoroughly referenced and builds on existing scholarship, I would suggest adding a few more texts to make the picture more complete. With regard to Everett criticism, I wondered if Danielle Morgan’s Laughing to Keep from Dying: African American Satire in the Twenty-First Century and Lisa A. Guerrero's Crazy Funny: Popular Black Satire and the Methods of Madness might not relate to the project at hand, even if they obviously deal with earlier novels; if the connection is remote, at least acknowledging them in a footnote seems appropriate. Besides that, the line about how "lynching is the symptom of a national disease" (6) made me wonder if Jacqueline Goldsby's A Spectacular Secret: Lynching in American Life and Literature might be a fruitful reference point to embed the novel more firmly in this larger context of violence and aesthetics (as Goldsby also argues that lynching wasn't just a Southern thing that the North could dismiss as alien but rather indeed a symptom of a national disease that only didn't break out in every place of the nation).

Again, these are very optional suggestions that would make a very good piece even stronger in my opinion; it is a fine contribution to Everett scholarship as it is, though, and I thoroughly enjoyed reading it.

Author Response

I have tried to streamline and focus my article through thorough revisions of the Introduction, and especially of the third part, which was pointed as the least convincing. I have tried to improve the clarity of the style, including by doing away with my first-person interventions. I was able to include, albeit briefly Danielle Morgan's book, whih I had read in the interim, but I will read Guerrro's soon, to help me with other projects.

Thank you so much for your thorough and sympathetic reading.